# An integrated proteome and transcriptome of B cell maturation defines poised activation states of transitional and mature B cells

Fiamma Salerno [1] ✉, Andrew J. M. Howden [2], Louise S. Matheson [1], Özge Gizlenci [1], Michael Screen[1], Holger Lingel [3], Monika C. Brunner-Weinzierl [3] & Martin Turner [1] ✉

During B cell maturation, transitional and mature B cells acquire cell-intrinsic features that determine their ability to exit quiescence and mount effective immune responses. Here we use label-free proteomics to quantify the proteome of B cell subsets from the mouse spleen and map the differential expression of environmental sensing, transcription, and translation initiation factors that define cellular identity and function. Cross-examination of the full-length transcriptome and proteome identifies mRNAs related to B cell activation and antibody secretion that are not accompanied by detection of the encoded proteins. In addition, proteomic data further suggests that the translational repressor PDCD4 restrains B cell responses, in particular those from marginal zone B cells, to a T-cell independent antigen. In summary, our molecular characterization of B cell maturation presents a valuable resource to further explore the mechanisms underpinning the specialized functions of B cell subsets, and suggest the presence of 'poised' mRNAs that enable expedited B cell responses.

In adult mammals B cell development is a multi-step process that starts in the bone marrow (BM) and continues in the spleen, where immature B cells form a transitional B cell continuum[1]. Transitional (T) B cells represent an important checkpoint for peripheral tolerance. The strength of B cell receptor (BCR) ligation in combination with secondary signals[2], determine whether T1 and T2 cells die or differentiate to mature marginal zone (MZ) or follicular (FoB) B cells[3,4]. A T3 transitional stage consists of anergic B cells rather than representing a developmental intermediate[5]. The process of peripheral maturation is thus key to ensure the generation of a highly diverse and protective B cell repertoire, while avoiding pathogenic reactivity.

Peripheral B cell subsets are characterized by distinct localization, migratory capacity and function. After passing a chemokine- and integrin-mediated checkpoint on survival and migration[6], T1 cells enter the splenic follicles and acquire expression of cell surface IgD, CD23 and CD21 to differentiate to T2 cells. T1 and T2 cells are defined by their

short half-life and propensity to undergo apoptosis upon crosslinking of surface IgM[7]. However, the ability of T1 cells to recirculate, recognise both self and foreign antigens and constitutively express low amounts of activation-induced deaminase (AID)[8] poises them for rapid responsiveness and class switch recombination. Thus, although they can contribute to the development of autoimmune diseases due to their enrichment for self-reactive specificities[9,10], T1 cells can also provide protection against infections[11].

FoB cells represent the vast majority of mature B cells and characteristically recirculate between lymphoid tissues. In contrast, MZ B cells represent only ~5–10% of mature B cells and reside in close proximity of the marginal sinus of the spleen, where incoming is filtered. Both subsets are relatively long-lived compared to transitional B cells and their best-characterised function is to mediate antibody production. However, their localization and cell-intrinsic features strongly influence the timing and quality of their responses[7,12,13]. Due to

[1]Immunology programme, The Babraham Institute, Cambridge, UK. [2]Cell Signalling and Immunology, University of Dundee, Dundee, UK. [3]Department of Experimental Pediatrics, Otto-von-Guericke-University, Magdeburg, Germany. ✉e-mail: fiamma.salerno@babraham.ac.uk; martin.turner@babraham.ac.uk

their unique position, MZ B cells are continuously exposed to blood-borne antigens and are poised for rapid differentiation to plasmablasts, thereby providing a first protective wave of antibodies[14]. Although FoB cells can also be engaged in early antibody production through the extrafollicular response, their collaboration with CD4[+] T-helper cells leads to the germinal centre reaction and gives rise to memory B cells and high-affinity long-lived plasma cells[15]. The molecular mechanisms that imprint the differential ability of transitional and mature B cells to exit quiescence and mount effective immune responses remain unresolved.

Systematic analysis of B cell transcriptomes has provided important insights into B cell identity[16–19]. However, studies performed in CD4[+] and CD8[+] T cells and in cell lines have demonstrated that for a substantial fraction of genes, transcriptome analysis is insufficient to predict protein content and therefore has limited ability to predict cellular behaviour[20–25]. This discordance is due to processes beyond those determining transcript abundance—such as translation efficiency and protein stability—that control protein amounts[26–28]. The current lack of quantitative proteomic data of primary B cells thus represents a major impediment for comprehensive understanding of B cell identity and responsiveness.

Here, we use label-free proteomics to analyse how the proteome of murine B cells is remodelled during peripheral maturation. This open-access data resource can be readily interrogated online via ImmPRes[29] (http://immpres.co.uk) and shows how environmental sensing, transcription factors and regulation of translation initiation control the identity and function of transitional and mature B cells. By quantifying protein copy numbers and the stoichiometry of protein complexes, we identify the translational repressor PDCD4 as a novel immune regulator that restrains B cell responses, in particular those of MZ B cells, to a T-cell independent antigen. In addition, by comparing the proteome and transcriptome within the same sample, we reveal how transitional and mature B cells maintain their phenotypic features and quiescent state while being poised for effective immune responses. Specifically, we discover a reservoir of protein coding mRNAs—which we call "poised" mRNAs—that are expressed by B cells without detectable protein, and might enable rapid protein production following activation. Altogether, our study highlights the predictive power of comparative proteomic and transcriptomic analysis and provides a valuable resource to enhance our molecular understanding of how immune regulators control B cell function.

## Results

### Proteomic analysis of peripheral B cell maturation

To study how the B cell proteome is remodelled during peripheral maturation, we sorted splenic T1, T2, MZ and FoB cells from C57BL/6 mice and performed quantitative label-free high-resolution mass spectrometry (MS). B cell subsets were sorted based on the expression of CD19, CD93, IgM, CD21 (also known as CR2) and CD23 (also known as FcER2) as depicted in Fig. 1a, and the expression profile of these markers was confirmed by MS (Supplementary Fig 1a, b). We identified 7560 protein groups, within which multiple protein isoforms can be assigned (Supplementary Data 1a). About 90% protein groups were found in at least three of four biological replicates and more than 70% were identified with high accuracy (i.e. by more than 8 total unique peptides and a ratio of unique+razor peptides ≥0.75[30]; Supplementary Fig 1c; Supplementary Data 1a), thus indicating the robustness of our dataset. To estimate protein mass and copy number per cell we applied the "proteomic ruler", which uses the MS signal of histones as an internal standard[31] (Supplementary Data 1a). Histones generally contribute some of the most intense peptides detected by MS and their sum intensity can be used to accurately estimate total protein abundance per cell without relying on error-prone steps of cell counting or calculation of absolute protein concentration[31]. Protein copy numbers from each B cell subset showed a strong Pearson correlation

coefficient (0.83–0.9) between the four biological replicates (Supplementary Fig 1d). The total protein content was slightly higher in MZ B cells compared to the other B cell subsets (Supplementary Fig 1e). This correlated with the larger size of MZ B cells, as indicated by their greater forward light scatter area (Supplementary Fig 1f). However, the physical difference in size did not merely scale up the abundance of all proteins. When we calculated the intergroup differences across the four B cell subsets of 6753 protein groups that were found in at least three of four biological replicates and identified by at least one unique peptide, we observed that 4937 protein groups (73% of the proteome) did not change (Fig. 1b). This is a core proteome common to all four B cell subsets. Conversely, the abundance of 1816 protein groups (27% of the proteome) differed indicating proteome remodelling during peripheral B cell maturation (Fig. 1b).

The 1816 differentially expressed (DE) protein groups formed six unsupervised clusters that are characteristic of a specific maturation stage of B lymphocytes (Fig. 1c). Clusters 1 and 2 contain proteins enriched in immature transitional cells; cluster 3 identified proteins that increased upon transition from T1 to T2 and maintained in mature B cells; cluster 4 included proteins that were mainly shared by T1 and MZ B cells; cluster 5 is characteristic of MZ B cells; whereas cluster 6 identified mature MZ and FoB cell stages.

The direct comparison of protein copy numbers between B cell subsets allowed us to determine proteins that were exclusive to a specific stage of B cell maturation and proteins that were expressed in two or more B cell subsets. All four maturation stages shared the expression of 1602 protein groups, yet they were found in different amounts (Fig. 1d and Supplementary Data 1b). T1 and MZ B cells represented the two extremes of this maturation process, as they were the only subsets that uniquely expressed 12 and 34 proteins, respectively (Fig. 1d and Supplementary Data 1c). For example, the transcription factor ZEB2, a novel regulator of B cell development[32], was exclusively found in T1 cells, whereas the key regulator of MZ B cell development NOTCH2 and its target DTX1[33] were uniquely found in MZ B cells. Conversely, T2 and FoB cells did not display a unique signature and largely shared their proteome between each other and with T1 and MZ B cells, thus supporting their intermediate stage during B cell differentiation.

### Transcription factors define B cell maturation stages

Amongst the 1816 DE proteins we surveyed transcription factors and found that their pattern of expression was characteristic of a specific B cell maturation stage (Fig. 2a), thus reflecting their role as regulators of B cell lineage commitment and identity. We estimated the amounts of key transcription factors per cell and observed a wide range in copy number. PAX5, that is indispensable in the establishment and maintenance of B cell identity[34], was abundant in all B cell subsets, with an average of 90,000 copies per cell (Fig. 2b). By contrast, ZEB2, TCF3 (E2A), and ARID3A, that characterize immature stages of B cell development[35–37], were mainly found in T1 cells and substantially less abundant than PAX5, with an average of 670, 11,000 and 20,000 copies per cell, respectively (Fig. 2c). The amount and expression pattern of these transcription factors might explain their relevance in mature B cell formation and function. In fact, whereas conditional deletion of PAX5 has deleterious effects on mature B cells[38], TCF3 and ARID3A control the formation of immature B cells but are dispensable for later stages of differentiation[36,37]. Also the transcription factor KLF2 that enforces a FoB cell phenotype[39], was found in low amounts in T1, T2 and FoB cells, and absent in MZ B cells (Fig. 2d). Conversely, MZ B cells expressed three to four-fold more of NFATC-1, -2 and -3 than FoB cells (Supplementary Fig 2a) consistent with their tendency to become more rapidly activated[40].

Upon encountering antigen, MZ B cells are known to rapidly differentiate into antibody-secreting cells[14]. Our proteomic data revealed that this may in part be attributed to the very low amounts of BACH2, a

transcription factor that suppresses plasma cell differentiation, and high amounts of IRF4 and ZBTB20, which promote plasma cell differentiation (Fig. 2e). Transcription factors characteristic of germinal centre B cells[41] were instead present in comparable amounts in all B cell subsets (Supplementary Fig 2b). MZ B cells also contained more STAT1 and STAT2 than other B cell populations (Fig. 2f). STAT1 promotes plasma cell differentiation in response to TLR- and type-I interferon-mediated stimulation[42], but the role of STAT2 in MZ B cell responses remains uncharacterized. By contrast, STAT4 was six times more abundant in FoB cells (18,000 copies/cell) compared to MZ B cells (3000 copies/cell) (Fig. 2f), yet it is still unclear whether STAT4 has a specific function in FoB cells. STAT5 and STAT6 that are activated

downstream of IL4 receptor were present in similar amounts in MZ and FoB cells (Supplementary Fig 2c). Taken together, these findings indicate that the distinct responsiveness of B cell subsets are, to some extent, pre-programmed by differential transcription factor expression. In addition, our analysis identified transcription factors that have been so far neglected in B cell biology, such as ZEB2, STAT2 and STAT4, thus opening new lines of inquiry.

**Environmental-sensing pathways are enriched in MZ B cells**
The profile of sensory and chemotactic receptors that are DE amongst splenic B cell subsets also highlighted clusters that are characteristic to each B cell subset (Fig. 3a). MZ B cells were notably enriched for the

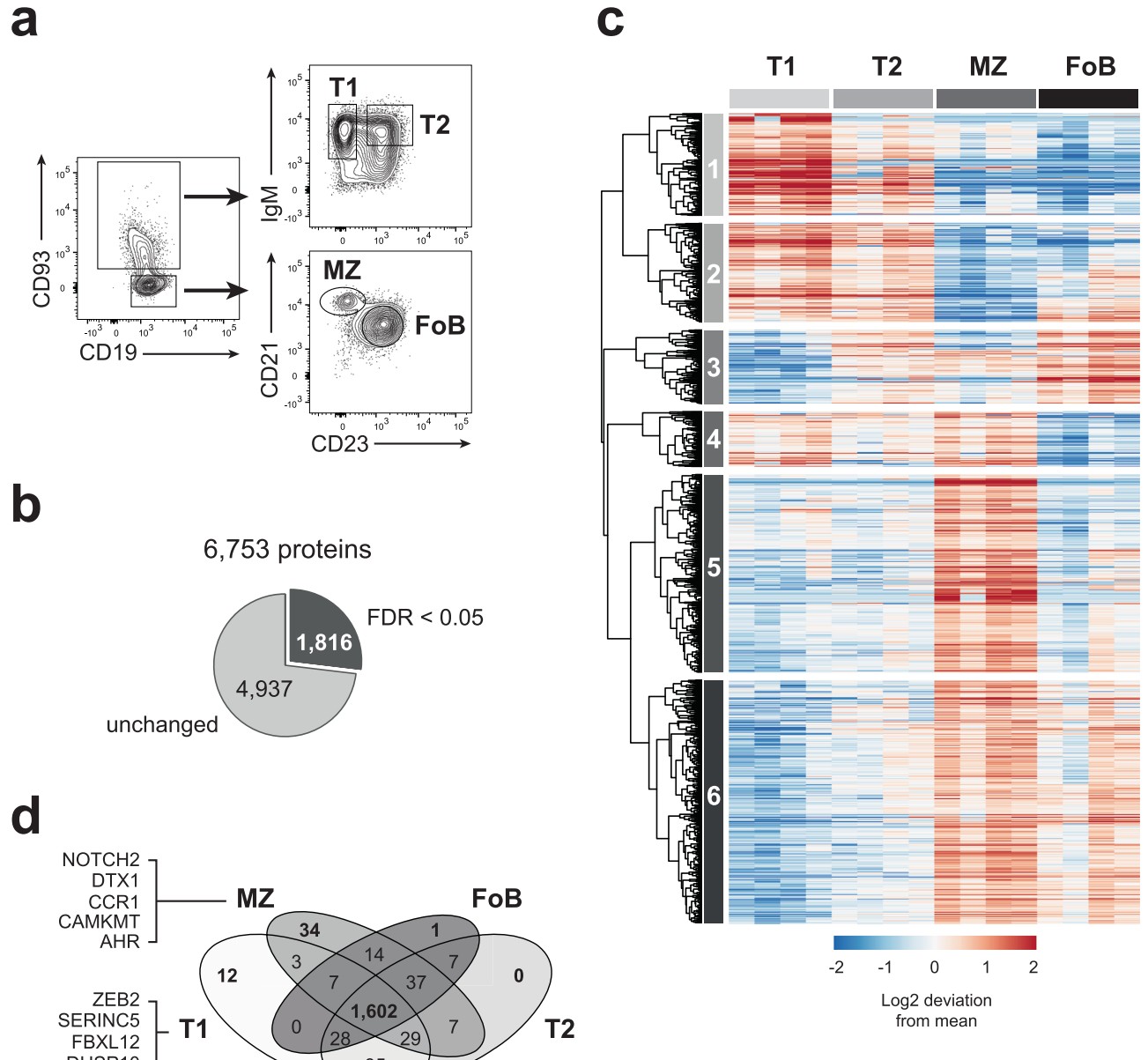

**Fig. 1 | Differential expression of cellular protein signatures during peripheral B cell maturation. a** Representative flow cytometry plots identifying T1 (CD19+ CD93+ IgM+ CD23-), T2 (CD19+ CD93+ IgM+ CD23+), MZ (CD19+ CD93- CD21+ CD23-) and FoB (CD19+ CD93- CD21- CD23+) cells. Full gating strategy is reported in Supplementary Fig 9a. **b** Proportion of proteins that were unchanged or differentially expressed within T1, T2, MZ and FoB cells based on ANOVA test followed by a Benjamini-Hochberg multiple testing correction with FDR < 0.05. Numbers indicate proteins that were identified in at least three out of four biological replicates and by

more than one unique peptide. The full list of proteins is provided in Supplementary Data 1a. **c** Heat map showing the log2-fold deviation from the mean of normalized copy numbers of 1816 DE proteins as in **b**. **d** Venn diagram indicates number of proteins that were found in one or more B cell subsets based on their copy numbers. Example of proteins that were exclusively found in T1 or MZ cells are reported on the left of the diagram. All proteins are listed in Supplementary Data 1b, c. In **b**–**d**, B cells were sorted from n = 4 biologically independent replicates from two independent experiments. Source data of (**c**) are provided as Source Data file.

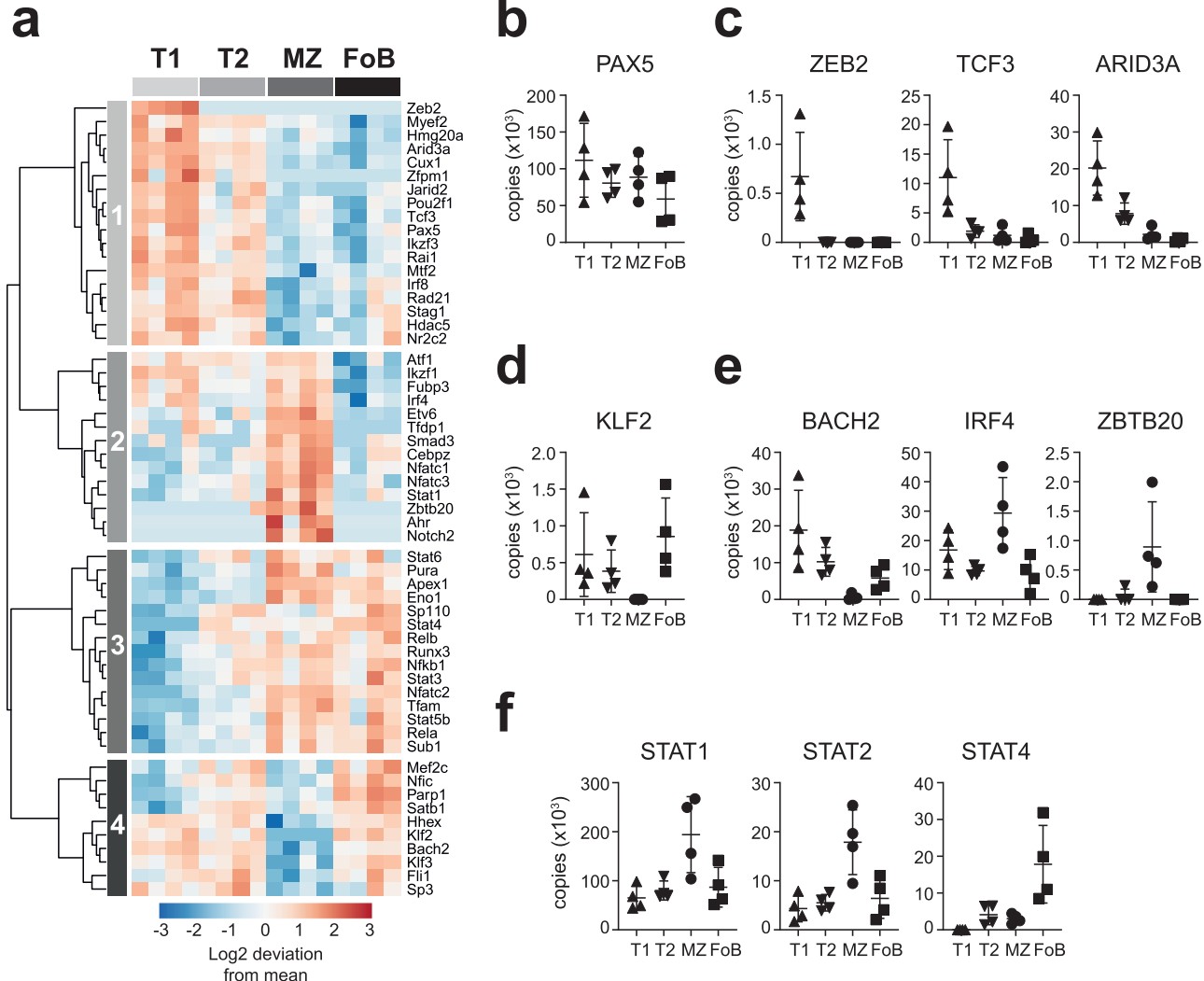

**Fig. 2 | Expression profile of transcription factors defining B cell maturation stages. a** Heat map depicts the log2-fold deviation from the mean of normalized copy numbers of 57 transcription factors that were differentially expressed among T1, T2, MZ and FoB cells (FDR-adjusted *p* values < 0.05, calculated as in Fig. 1b). **b–f** Graphs show copy numbers per cell of transcription factors that were selected based on their known relevance to immunity (*n* = 4 mice; mean ± SD). Source data are provided as Source Data file.

receptors sensing pathogen-associated molecular patterns. They preferentially express toll-like receptors TLR3 and TLR7, and the nucleotide-binding oligomerization domain (NOD)-like receptors NLRC4 and NLRX1 (Fig. 3a, b), which enable their immediate response to insults. Conversely, TLR9 that provides costimulation for the B cell response to antigen[43] and contributes to peripheral tolerance[44], was equally expressed within all B cell maturation stages (Fig. 3b). In association with TLR expression, MZ B cells displayed greater amounts of CD180 and TACI (also known as TNFRSF13B or CD267) (Fig. 3c) that synergize with TLR signalling and enhance antibody responses[45,46]. The abundance of CD180 and TACI was confirmed by surface antibody staining and flow cytometry (Fig. 3c). Whereas sensors for pathogen recognition were preferentially expressed by MZ B cells, the costimulatory receptor CD40 and the surface Ig-associated proteins CD79a/CD79b were similarly abundant on MZ and FoB cells (Fig. 3d and Supplementary Fig 3a). Likewise, the copy numbers of the inhibitory molecules CD22, CD72 and SiglecG were not different between the two mature B cell subsets (Fig. 3d and Supplementary Fig 3b). These data suggest that both MZ and FoB cells have the potential to respond to T cell-dependent antigens. However, the differential expression of sensory receptors poise MZ B cells to respond to a more diverse set of antigenic stimuli compared to FoB cells.

The sensing and response to the environment also relies on metabolite transporters and pathways that process them following their uptake. Whereas the glucose transporter SLC2A1/GLUT1 was found in equivalent amounts in all peripheral B cell subsets, SLC2A3/GLUT3, which is known to have higher glucose transport capacity than GLUT1[47], was more abundant in MZ B cells (Fig. 3e). Together with an increase in glycolytic protein content (Supplementary Fig 3c), our data indicated that basic glycolytic flux is likely to be the greatest in MZ B cells. High metabolic activity has been associated with transitional B cells and the acquisition of metabolic quiescence found to be a requirement for maturation to FoB cells[48]. We therefore questioned whether different metabolic signatures distinguish MZ and FoB cells. We inferred the potential for peripheral B cell subsets to uptake lactate, amino acids and lipids by considering the expression of solute transporters. We found that the lactate/pyruvate transporter SLC16A1, and the amino acid transporters SLC1A5 and SLC7A5, were mainly expressed by T1 cells, and less by mature B cells (Fig. 3e). Also, the expression of the fatty acid transporters SLC27A1 and FABP5 were greater in transitional B cells compared to mature B cells (Fig. 3e). However, the expression of some fatty acid transporters was subset-specific, as CD36 was approximatively 6-fold higher and 18-fold higher on MZ B cells compared to T1/T2 and FoB, respectively (Fig. 3e), while

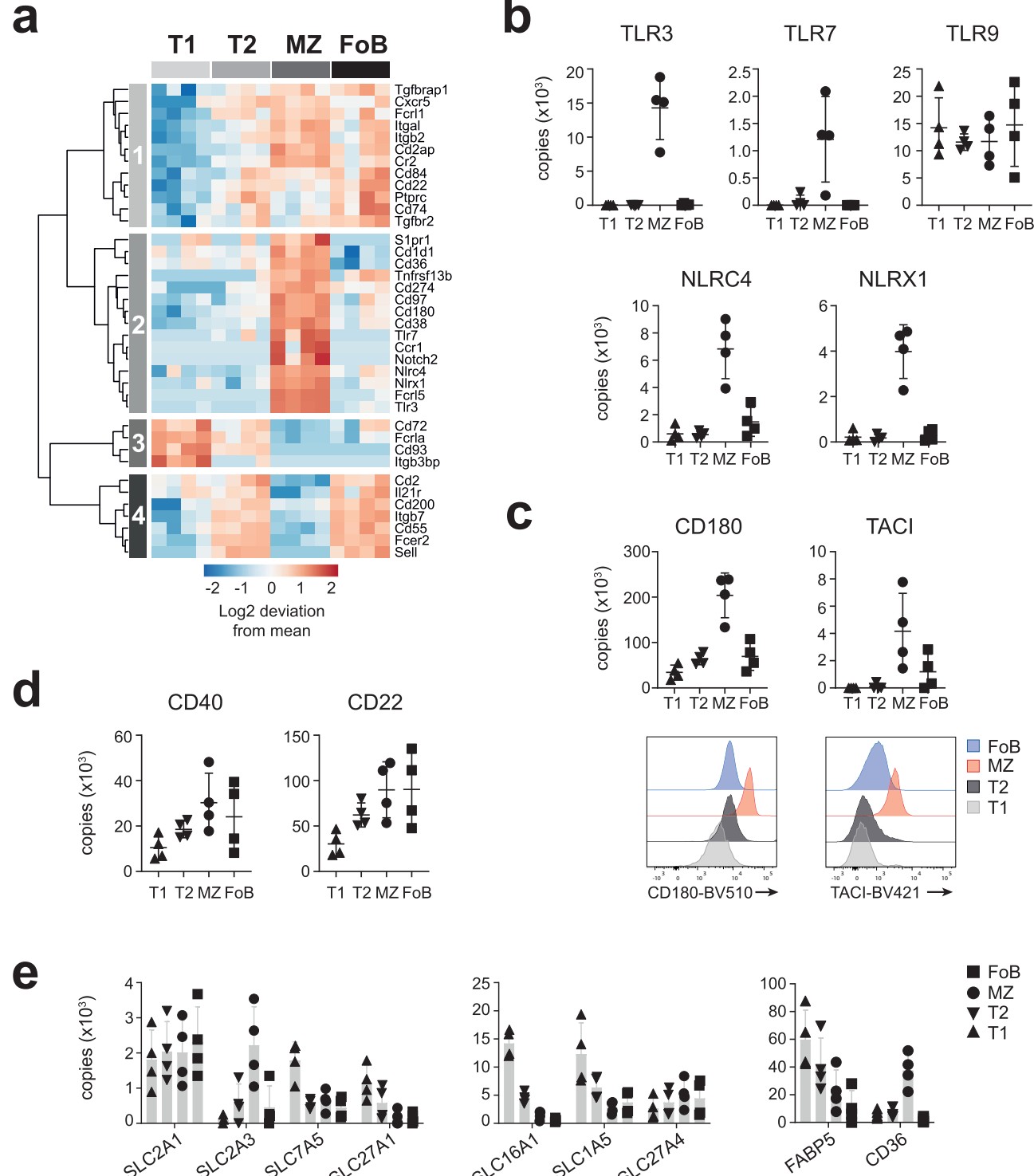

**Fig. 3 | Receptors involved in environmental-sensing and nutrient transport in B cells. a** Heat map shows the log2-fold deviation from the mean of normalized copy numbers of 38 receptors that were differentially expressed among T1, T2, MZ and FoB cells (FDR-adjusted *p* values < 0.05, calculated as in Fig. 1b). The list of receptors was manually curated and includes CD molecules; toll-like receptors; NOD-like receptors; Fc receptors; complement receptors; TNF receptor super-family; receptors for cytokines such as interleukins, interferons, TGF-β; chemotactic receptors belonging to the CCR, CXCR or sphingosine-1-phosphate families; adhesion molecules; ADAM metalloproteinases and Notch receptors. **b**–**d** Graphs display copy numbers per cell of receptors selected based on their biological function. In **c**, histograms represent CD180 and TACI expression as detected by flow cytometry. **e**, Copy numbers of the major glucose-, lactate- and amino acid-transporters and selected fatty acid transporters (*n* = 4 mice; mean ± SD). Source data are provided as Source Data file.

SLC27A4 was equally abundant on both mature B cell subsets. Of note, CD36 has a dual role and its function as scavenger receptor for pathogens could further explain its preferential expression on MZ B cells[49]. Together, our data indicate that MZ B cells differ from metabolically quiescent FoB cells, as they maintain a proteome better able to engage anabolic metabolism and to rapidly respond to environmental changes.

## Gene-specific divergence of protein and mRNA abundance in B cell subsets

Transcriptomic analysis has been instrumental to advance our understanding of B cell development and function. However, for some genes mRNA amounts are unable to predict protein amounts[20,21,25,28] and the correlation between transcript and protein abundance in B cells has yet to be examined.

To assess the correlation between the proteome and transcriptome of peripheral B cell subsets, we directly compared protein copy numbers to transcripts per million (TPM) determined by Illumina sequencing within the same biological samples. We restricted our analysis to 7303 genes that were identified both by RNA-sequencing and proteomics in at least one population (Supplementary Data 2), thus including 96.6% of detected proteins and 47.8% of identified transcripts. For this analysis we exclusively selected the first protein identifier for each protein group, which corresponds to the most represented protein isoform (i.e. identified by the highest number of peptides). The amounts of individual proteins displayed a wider dynamic range than that of mRNAs; whereas proteins spanned more than seven-orders of magnitude, mRNAs spanned only five-orders of magnitude (Supplementary Fig 4a). The Pearson correlation (r) of log-transformed TPM and protein copy numbers ranged between 0.43 and 0.49 in T1, T2, MZ and FoB cells (Supplementary Fig 4b). Although it is difficult to estimate the contribution of measurement noise[24,50], this result suggests a moderate correlation within the variance in mRNA and protein amounts across the entire dataset. The mRNA/protein correlation only slightly increased when using the non-parametric Spearman ranking method ($\rho$ = 0.45–0.49; Supplementary Fig 4b) and was similar to previously published estimates in bacteria and eukaryotes (summarized in refs. 22,23), indicating that the impact of translational and post-translational processes on protein abundance is a general feature that is independent of a specific cell type or differentiation stage.

This result prompted us to query firstly whether the relationship between mRNA and protein abundance was equal for gene-clusters selected based on their biological function; and secondly how this related to within-gene analysis, i.e. how mRNA and protein abundance of a specific gene changes across the four B cell subsets. To answer to these questions, we studied genes that undergo rapid on/off switching upon activation and determined where their mRNA/protein expression was situated in relation to the overall across-gene trend. This is indicated by a line that was calculated based on density contours and by assuming a positive relationship between mRNA and protein expression (Supplementary Fig 4c). We considered first mRNAs containing a 5′-terminal oligopyrimidine (TOP) motif that encode mostly translation factors and ribosomal proteins[51,52], and were found to be translationally repressed in naive T cells[28]. We found that T1, T2, MZ and FoB cells expressed uniformly high amounts of 113 out of 123 TOP-containing genes both at the mRNA and protein level, with most genes lying close to the trend line. This result thus indicates a positive mRNA/protein relationship and suggests that translation of TOP-mRNAs is not repressed in transitional and mature B cells (Fig. 4a).

We then assessed the relationship between mRNA and protein for genes encoding 265 transcription factors. Although many of these genes were close to the trend line at all maturation stages (e.g. for PAX5, IRF4, BACH2), transcription factors displayed greater variability compared to TOP-mRNAs and some were exclusively detected as

mRNA, but not protein (Fig. 4b). For example, while the ZEB2, KLF2 and ZBTB20 proteins were strictly associated with a specific B cell maturation stage, all B cell subsets expressed comparable levels of their encoding mRNAs (Fig. 4b). In particular, ZEB2 protein was restricted to T1 cells, yet T2, MZ and FoB cells retained the expression of ~40 TPM of *Zeb2* mRNA. Similarly, MZ B cells contained ~300 TPM of *Klf2* mRNA, whereas KLF2 protein was undetectable (Fig. 4b). In contrast to TOP-mRNAs and transcription factors, genes encoding sensory and chemotactic receptors (as identified in Fig. 3) were overall lower compared to the trend line. Moreover, also in this gene-group we found genes that were detected as mRNA, but not protein (Fig. 4c, d). For example, all B cell subsets expressed ~400 TPM of *Tnfrsf13b* mRNA whereas its protein (TACI) was detected exclusively in MZ and FoB cells. Additionally, MZ and FoB cells both express ~100 TPM of *Notch2* mRNA, yet NOTCH2 was found exclusively in MZ B cells (Fig. 4d). Collectively, these data indicate that although the overall mRNA abundance correlates moderately with protein amounts in B cells, a positive mRNA/protein relationship becomes evident when interrogating specific gene-groups. In addition, within-gene analysis identified individual transcripts that show evidence of translational control dependent on a specific B cell differentiation stage.

## Transitional and mature B cells express a poised gene-signature

B cells undergo basal transcription, which has been proposed to poise the genome for increased transcription following stimulation[19]. In T cells and cells of the innate immune system, some mRNAs associated with early activation and effector function, termed "poised" mRNAs, accumulate but are not translated[27,28,53–55]. Here we tested the hypothesis that peripheral B cell subsets express poised mRNAs to enable rapid activation and differentiation into antibody-secreting plasmablasts (PB). We compared the transcriptome of T1, T2, MZ, and FoB cells to 1) a list of genes that were selected based on their increased expression upon 2 h of B cell stimulation with anti-IgM or LPS[56], which we refer to as "early activation genes" (Supplementary Data 3a); and 2) a list of genes that were differentially expressed in immature (B220+ Blimp1+)-PB and mature (B220- Blimp1+)-PB compared to FoB cells[16], which we refer to as "PB-related genes" (Supplementary Data 3b; Supplementary Fig 5a). We found that 412 early activation genes and 1007 PB-related genes were expressed at the mRNA level in T1, T2, MZ or FoB cells prior to activation, and were thus candidates to be poised mRNAs.

To identify protein-coding genes that were expressed at the mRNA but not at the protein level, we sought to compare our transcriptomic and proteomic datasets. To do this, we had to consider the different sensitivity of RNA-seq and mass spectrometry, as the greater sensitivity of RNA-seq may lead to misidentification of poised transcripts due to stochastic detection of very low abundance transcripts. To overcome this issue, we first calculated the probability to detect proteins for defined amounts of protein-coding mRNAs. This related RNA and protein levels of all detected protein-coding mRNAs per population, and then calculated the probability to detect protein for each bin. A minimum of 10 TPM gives a 50% chance of finding the corresponding protein. Conversely, mRNA with TPM < 10 is associated with less than 20–30% chance, indicating an unacceptably high probability of classifying these low abundant mRNAs as poised mRNAs, when in reality they are not (Supplementary Fig 5b). We therefore restricted our analysis to between 212–243 early activation genes and 318–375 PB-related genes with TPM > 10 in each B cell subset. The abundance of these selected genes is between 10 and ~3500 TPM (Supplementary Fig 5b).

Next, to assess whether these transcripts were present as full-length protein-coding mRNAs, we performed Oxford Nanopore Technology (ONT) sequencing of the same samples. We found that about 80% of both early activation and PB-related genes were full-length mRNAs (Supplementary Fig 5c) and their detection by ONT was

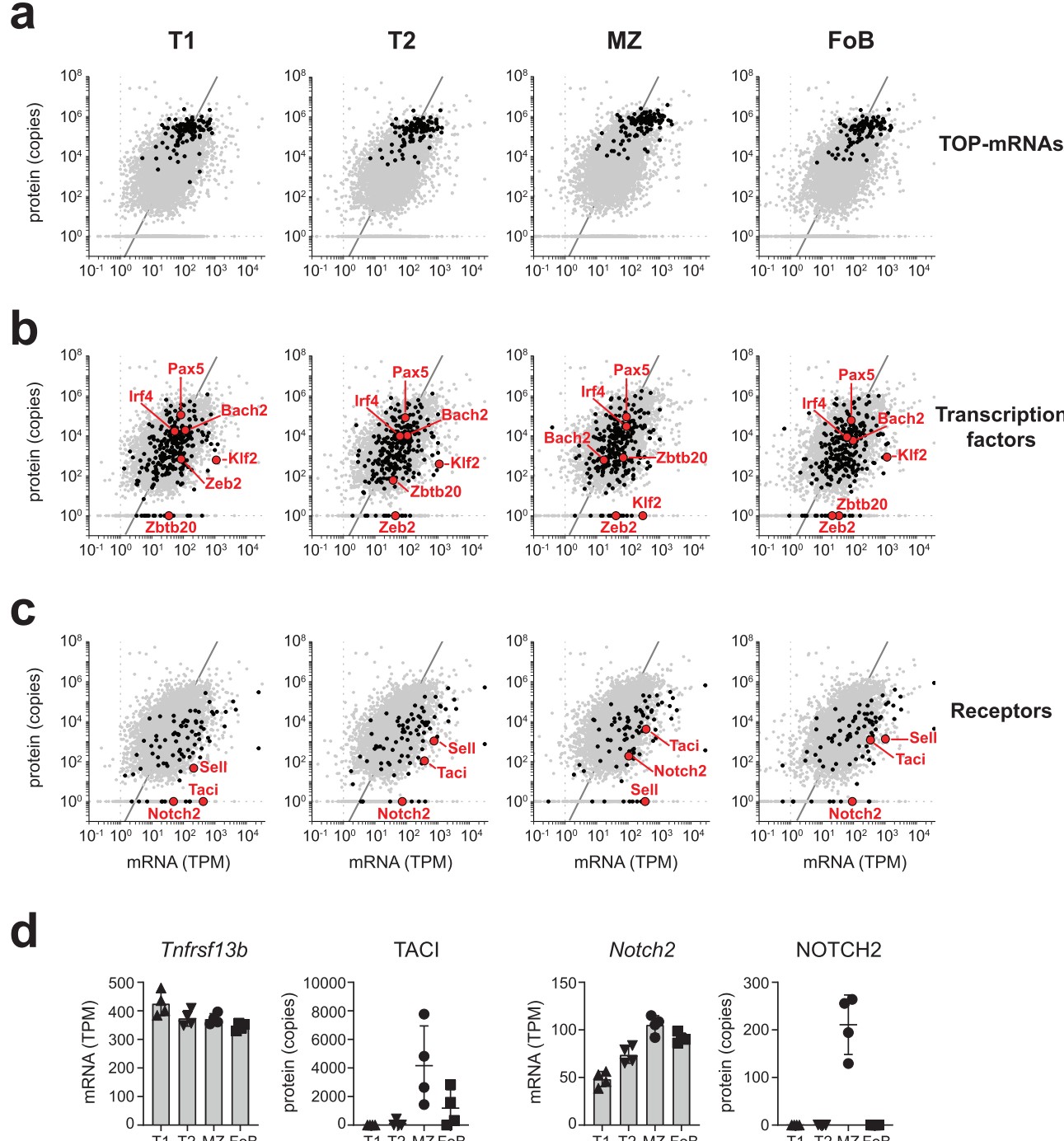

**Fig. 4 | Integration of mRNA and protein abundance during B cell development.**
**a–c** Scatter plots show the average mRNA abundance (in transcripts per million - TPM) and the average protein abundance (in copy numbers) of T1, T2, MZ and FoB cells. Expression and trend line of the whole dataset encompassing 7303 genes is reported in each graph (grey dots and line). Black dots indicate expression of 113 TOP-mRNAs (**a**), 265 transcription factors (**b**), and 74 sensory and chemotactic receptors (**c**). Red points highlight specific genes of interest within those categories. A list of TOP-mRNAs was extracted from refs. 51,52; transcription factors were annotated from Gene Ontology term 0003700; a list of receptors was manually curated as in Fig. 3. **d** mRNA and protein abundance of *Tnfrsf13b*/TACI and NOTCH2 in B cell subsets (*n* = 4 mice; mean ± SD). The full list of TPMs and copy numbers is provided in Supplementary Data 2. Source data of (**d**) are provided as Source Data file.

independent of their relative abundance (Supplementary Data 4 reports a side-by-side comparison of TPMs of genes that were detected by both ONT and Illumina sequencing). We then calculated how many of these genes were detected at the protein level in our proteomic dataset. As a control for protein detection, we generated expression-matched gene sets for the 189–221 early activation and 295–345 PB-related genes detected as full-length mRNAs, by randomly selecting

100 protein-coding genes with the closest expression for each mRNA. We calculated protein detection within each B cell subset and found that the number of proteins detected within the early activation gene list was about 30% lower than control genes (Supplementary Fig 5d). This indicates that T1, T2, MZ and FoB cells are enriched for mRNAs with critical roles in early activation that are expressed without detectable protein. We classify these transcripts as a poised early

activation signature (Supplementary Fig 5e; Supplementary Data 5a). For example, transitional and mature B cells expressed ~200 TPM and ~1000 TPM of mRNA encoding the early activation marker CD69 and the immediate early genes *Fos*, which were identified as full-length transcripts by ONT sequencing (Fig. 5a, b, Supplementary Fig 5f). Whereas we did not detect CD69 or FOS in our proteomic data, 100 expression-matched mRNA controls had a median of at least 12,000 associated protein copies per cell (Fig. 5c, Supplementary Fig 5f). Similarly, T2, MZ and FoB cells expressed ~85 TPM of poised *Nr4a1* mRNA (encoding NUR77), however NUR77 is detectable only 2 h after antigenic stimulation[57] (Fig. 5d-f). mRNA encoding the transcription factor c-Myc, which is important for activation, proliferation and differentiation of B cells[58,59], was also poised in all B cell subsets, but particularly abundant in MZ B cells, in agreement with their predisposition to rapid responses (Fig. 5g–i).

MZ B cells also exhibit a small enrichment for poised mRNAs with roles in generating antibody-secreting PB (Supplementary Fig 6a, b, Supplementary Data 5b). This was also evident in T1, but not in T2 or FoB cells, for which the overall number of detected proteins was within the range of what is expected by chance (Supplementary Fig 6a). Because rapid differentiation of MZ B cells to PB is at least in part due to the persistent expression of unfolded protein response (UPR)-related mRNAs[16], we examined the abundance of individual UPR-related transcripts within our identified poised signature and found that this feature was not restricted to MZ B cells, but shared with T1, T2, and FoB cells. All B cell subsets expressed high amounts of unspliced *Xbp1* mRNA without detectable protein (Fig. 6a–c). Moreover, mRNAs encoding the transcriptional activator ATF4, the UPR-sensors ATF6, and the gene associated to ER expansion EDEM1 were also present as poised mRNAs (Fig. 6d–h, Supplementary Fig 6c). Together, our data show that transitional and mature B cells express a poised gene-signature encoding early activation and PB-related genes (Supplementary Data 5a, b), which may facilitate activation and antibody secretion.

### Regulators of mRNA translation in transitional and mature B cells

We next sought to identify the factors that regulate mRNA translation during specific B cell maturation stages. We calculated the total ribosomal protein content based on copy numbers of ribosomal subunits, and found that it represents 4.6% of the MZ B cell proteome versus 3.5% of the T1, T2 or FoB cell proteome (Fig. 7a). Similarly, the translation initiation complex, including the components of the eukaryotic initiation factor 4 F complex (i.e., eIF4A1, eIF4B, eIF4E and eIF4G paralogs), the components of the eIF3 complex, and the components of the eIF2 complex, which delivers the methionine-tRNA to 40 S ribosomal subunit, were also enriched in MZ B cells (Fig. 7b, c and Supplementary Fig 7a–c).

Next, we assessed the abundance of factors that inhibit translation initiation. A common mechanism of translation inhibition, especially during stress, is the phosphorylation of eIF2α, which inhibits the GTPase activity of eIF2B, thereby limiting the regeneration of the ternary complex[60]. This phosphorylation is mediated by the HRI, PKR, PERK and GCN2 kinases (also known as eIF2AK1, eIF2AK2, eIF2AK3, and eIF2AK4, respectively)[60], and the last three of these kinases were found in our proteomic datasets. Whereas PERK was detected in similar low amounts in all B cell populations, PKR and GCN2 were enriched in mature B cells (Supplementary Fig 7d). In particular, PKR was the most abundant of the three kinases and preferentially expressed in MZ B cells (~5000 copies/cell). Considering that PKR can also control transcription by NFκB activation[61], it would be interesting to further investigate its specific role in MZ B cells.

A distinct mechanism to limit cap-dependent translation is mediated by the action of the eIF4E-binding proteins (4E-BPs), which is relieved by activation of the mTOR pathway[62]. Of the three known eIF4E-BPs, only 4E-BP2 was detected in our proteomic analysis, although with low abundance and not in all biological replicates (Supplementary Fig 7e). These data suggest that 4E-BPs might not be influential in quiescent B cell subsets. Translation initiation may also be inhibited by PDCD4, which binds and sequesters two molecules of the RNA helicase eIF4A1[63–65]. PDCD4 was highly abundant in all peripheral B cell subsets and notably enriched in MZ B cells, which we confirmed by intracellular flow cytometry (Fig. 7d). We found that T1, T2 and FoB cells expressed eIF4A1 in large excess compared to PDCD4 (eIF4A1/PDCD4 ratio ~5), but in MZ B cells PDCD4 copies are sufficient to bind the majority of eIF4A1 (eIF4A1/PDCD4 ratio = 2.6; Fig. 7e). Although PDCD4 might act as a negative regulator of mRNA translation in all B cell subsets, the stoichiometry of its interaction with eIF4A1 indicates that it might have a specific role in MZ B cells.

### PDCD4 restrains MZ B cell responses

To study the role of PDCD4 in B cells in vivo, we established mixed chimeras by transferring 20% of either CD45.2+ wild-type (WT) or CD45.2+ PDCD4-deficient (PDCD4 KO) bone-marrow cells together with 80% of CD45.1+ μMT-deficient bone-marrow cells into lethally irradiated CD45.1+ B6.SJL recipient mice. The μMT mutation prevents the formation of mature B cells, so that all B cells in these chimeras will derive from CD45.2+ WT or PDCD4 KO stem cells, whereas all other hematopoietic lineages are primarily PDCD4-sufficient[66]. PDCD4 KO cells efficiently reconstituted the mature B cell pool in the spleen, however the frequency and numbers of PDCD4 KO B cells were overall lower comparing to WT B cells (Fig. 8a). In addition, PDCD4 KO μMT chimeras displayed reduced numbers of T1, T2 and FoB cells, but not of MZ B cells, which were comparable to WT μMT chimeras (Supplementary Fig 8a). Because of the difference in B cell numbers, we also generated competitive chimeras by transferring equal amounts of CD45.2+ WT and CD45.2+ PDCD4 KO bone-marrow cells together with CD45.1+ B6.SJL bone-marrow cells into lethally irradiated B6.SJL recipient mice. Following reconstitution, WT and PDCD4 KO competitive chimeras displayed similar numbers and proportions of T1, T2, MZ and FoB cells (Fig. 8b and Supplementary Fig 8b).

We next investigated the role of PDCD4 in B cell responses in vivo by administration of the T-cell independent antigen NP-Ficoll, in the absence of adjuvant, to the μMT chimeras and competitive chimeras. MZ B cells contribute substantially to the rapid extrafollicular IgG3 response to NP-ficoll, whereas the participation of FoB cells is minor[67]. To maximize the engagement of MZ B cells, we injected NP-Ficoll intravenously, thus facilitating its capture by MZ B cells that reside where incoming blood is filtered[68–70]. We analysed B cell responses seven days following immunization and found that the frequency of PDCD4 KO CD138+ TACI+ CD19int/low IgD- plasmablasts (PBs) was greater than WT PBs in both μMT and competitive chimeras (Fig. 8c). This indicates that PDCD4 limits the ability of B cells, and in part of MZ B cells, to differentiate to PBs or of PBs to accumulate and/or survive. Although the proportion of intracellular-IgM+ and intracellular-IgG3+ PBs was similar between PDCD4 KO and WT chimeras (Supplementary Fig 8c), the mean fluorescence intensity of NIP+ ic-IgM+ and NIP+ ic-IgG3+ staining was higher for PDCD4 KO PBs compared to WT PBs (Supplementary Fig 8d). Also, the frequency of NP-specific IgM and IgG3 antibody-secreting cells (ASC) measured by ELISpot was increased in the spleen of PDCD4 KO μMT chimeras compared to WT (Fig. 8d), indicating an enhanced B cell response in the absence of PDCD4. We next measured the titer of NP-specific IgM and IgG3 antibodies before and after NP-Ficoll immunization of μMT chimeras. PDCD4 KO B cells already produced higher amounts of NP-specific IgM prior to immunization. Elevated IgM production was maintained at day 4 after immunization, however WT B cells could match the production levels of PDCD4 KO B cells by day 7 (Fig. 8e). Conversely, WT and PDCD4 KO B cells exhibited similar kinetics of NP-specific IgG3

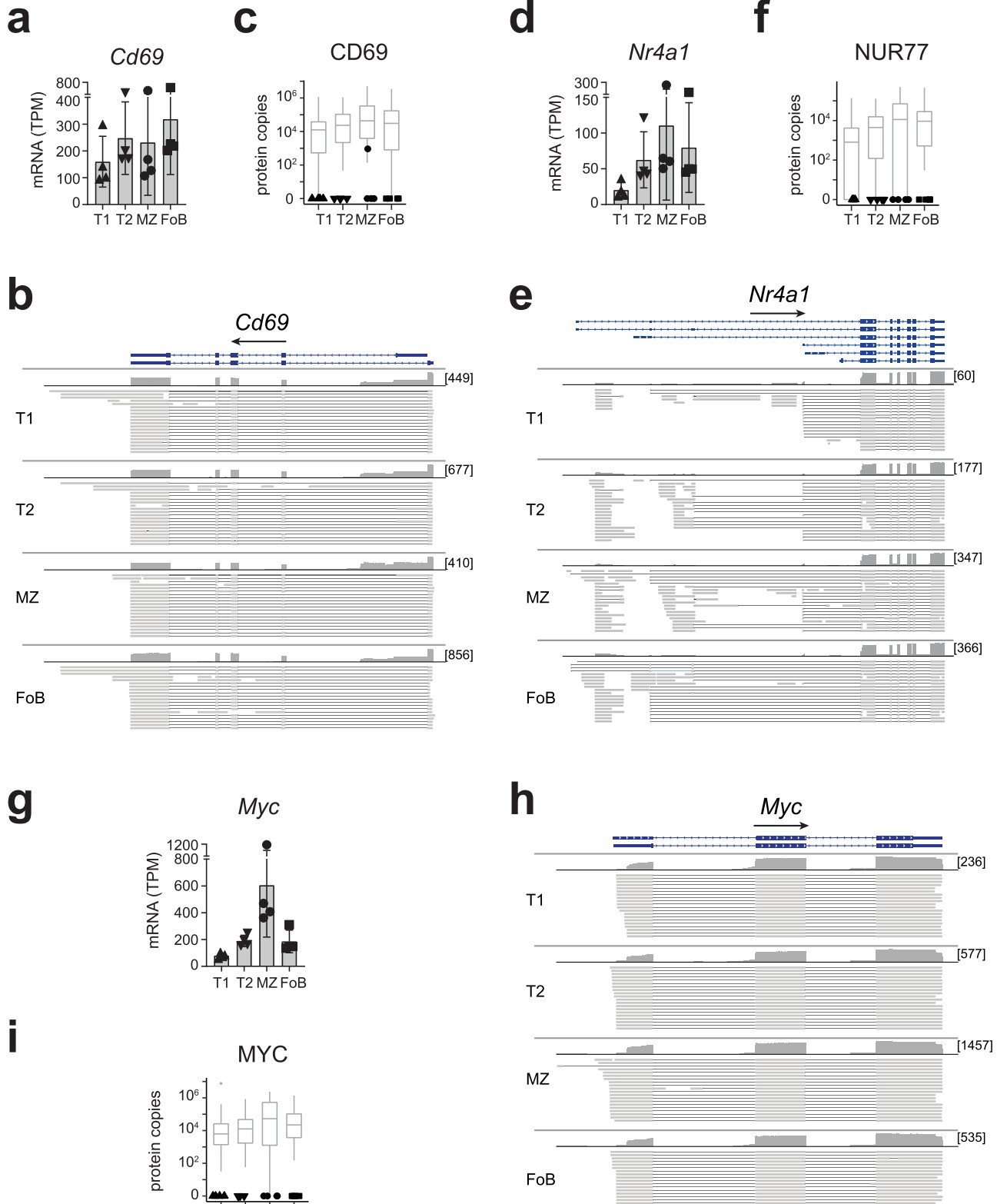

**Fig. 5 | B cells express early activation genes in a poised state. a, d, g** mRNA abundance in TPM of selected early activation genes as measured by Illumina sequencing (*n* = 4 mice; mean ± SD). Genome browser view displays individual long-reads from ONT sequencing of *Cd69* (chr6: 129,266,982-129,275,447) (**b**), *Nr4a1* (chr15:101,254,093-101,275,115) (**e**), *Myc* (chr15: 61,983,341-61,992,361) (**h**), visualized with the Integrative Genomics Viewer. RefSeq GRCm38 annotation (blue), coverage of (dark grey) and aligned long-reads (light grey) were reported for each gene. Lines connecting light grey boxes indicate splicing junctions between aligned sequences. In squared brackets the maximum read coverage for each B cell population. **c, f, i** Black symbols show protein detection of indicated genes within our proteomic dataset. Grey boxes represent protein copy numbers (mean ± SD) for the 100 protein-coding genes with closest RNA expression (log2 TPM) to the indicated gene. The centre and bounds of the boxes represent the median and interquartile range, respectively. The upper and lower whiskers extend to the maximum and minimum values, respectively, or, in cases where outliers are displayed, to the 75th/25th percentile +/− 1.5x interquartile range. Source data are provided as Source Data file.

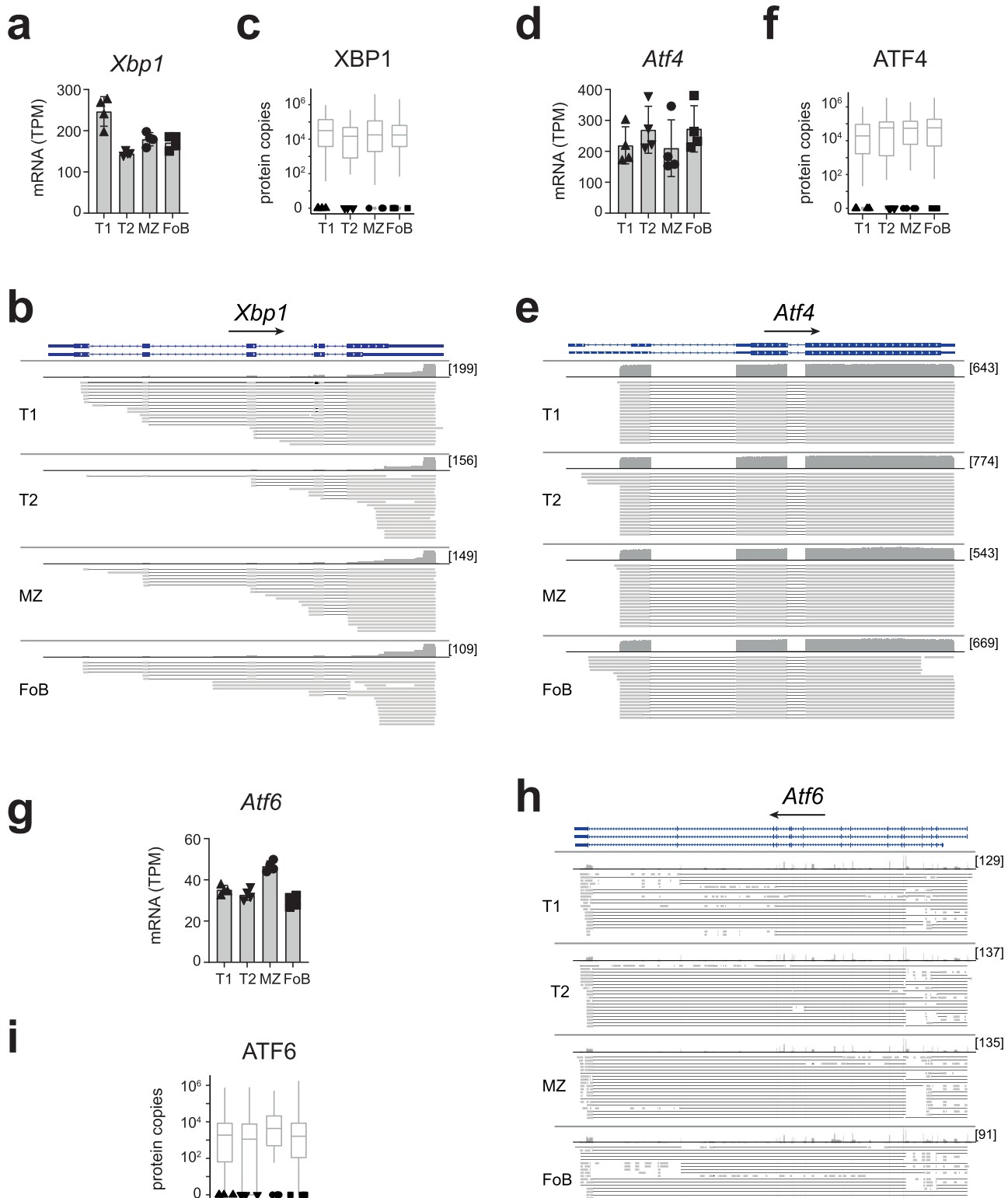

**Fig. 6 | B cells express PB-related genes in a poised state. a, d, g** mRNA abundance in TPM of selected PB-related genes as measured by Illumina sequencing (*n* = 4 mice; mean ± SD). Genome browser view displays individual long-reads from ONT sequencing of *Xbp1* (chr11: 5,520,014-5,526,248) (**b**), *Aft4* (chr15: 80,255,164-80,257,565) (**e**), *Atf6* (chr1:170,703,549-170,870,441) (**h**), visualized with the Integrative Genomics Viewer. RefSeq GRCm38 annotation (blue), coverage of (dark grey) and aligned long-reads (light grey) were reported for each gene. Lines connecting light grey boxes indicate splicing junctions between aligned sequences. In squared brackets the maximum read coverage for each B cell population. **c, f, i** Black symbols show protein detection of indicated genes within our proteomic dataset. Grey boxes represent protein copy numbers (mean ± SD) for the 100 protein-coding genes with closest RNA expression (log2 TPM) to the indicated gene. The centre and bounds of the boxes represent the median and interquartile range, respectively. The upper and lower whiskers extend to the maximum and minimum values, respectively, or, in cases where outliers are displayed, to the $75^{th}/25^{th}$ percentile +/− 1.5x interquartile range. Source data are provided as Source Data file.

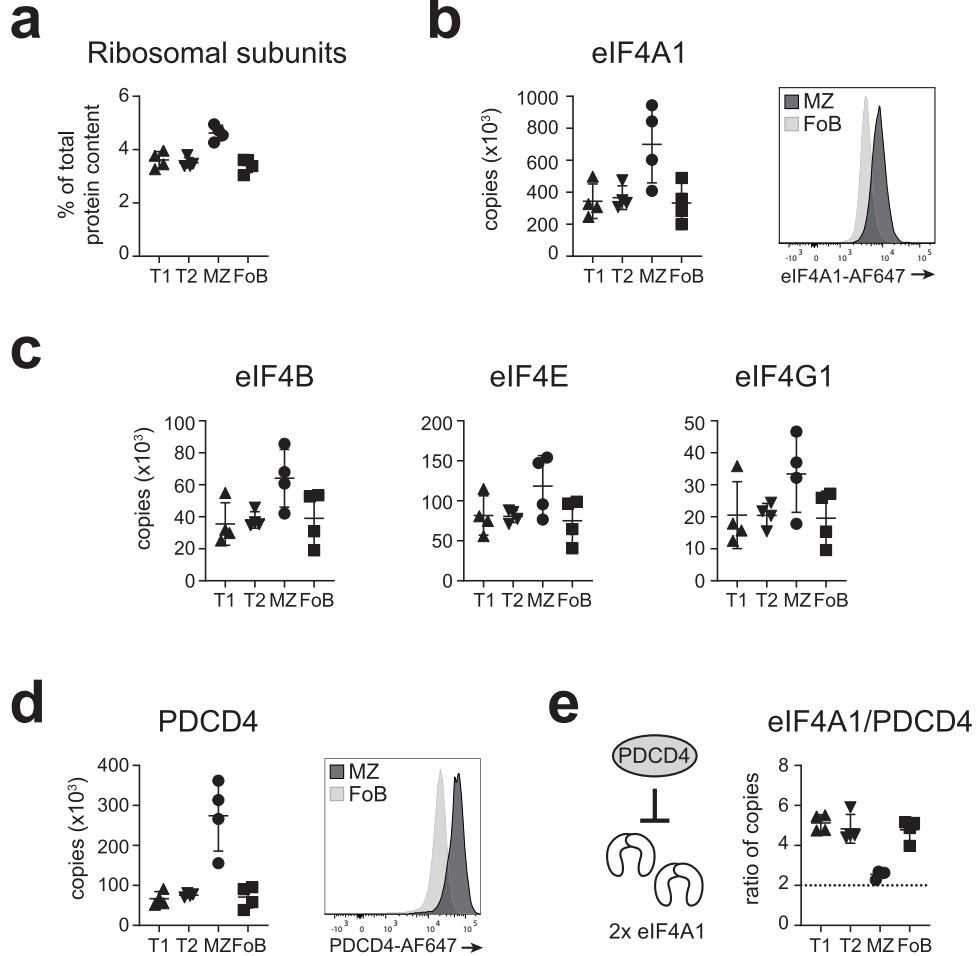

**Fig. 7 | Regulators of mRNA translation in B cells. a** Total content of ribosomal proteins was calculated as sum of CN x MW / $N_A$, where CN is protein copy number, MW is protein molecular weight, and $N_A$ is Avogadro's constant. Graph depicts ribosomal proteins as a proportion to total protein content in T1, T2, MZ and FoB cells. **b**, **c** Graphs display copy numbers of key components of the eIF4F mRNA translation initiation complex (eIF4A1, eIF4B, eIF4E, eIF4G1). Overlay histogram represents flow cytometric detection of eIF4A1 in MZ and FoB cells. **d** PDCD4 protein amounts detected by mass spectrometry (left) or flow cytometry (right). **e** Ratio between EIF4A1 and PDCD4 copy numbers was calculated in each B cell subset. Graphs display mean ± SD (*n* = 4 mice). Source data are provided as Source Data file.

production, yet PDCD4 KO B cells released greater amounts of IgG3 antibodies by day 7 post-immunization (Fig. 8e).

The enhanced response of PDCD4 KO B cells to NP-Ficoll immunization supports the hypothesis that PDCD4 is a newly identified regulator of MZ B cells. To determine whether PDCD4 restrains global RNA translation, we measured puromycin incorporation into MZ B cells in vivo. We found that in the resting state WT and PDCD4 KO MZ B cells incorporated similar amounts of puromycin (Fig. 8f), suggesting that the effect of PDCD4 on translation was limited and did not affect translation globally. To quantify this effect, we examined how the loss of PDCD4 impacts the proteome of MZ B cells by performing label-free proteomics of WT and PDCD4 KO MZ B cells derived from unimmunized chimeric mice. We found that PDCD4 controls the expression of 136 proteins, of which 24 proteins showed a log2 fold change greater than 5 in PDCD4 KO MZ B cells (Fig. 8g). None of these proteins were included in the poised mRNA signature we identified, but comprised enhancers of BCR and TLR signalling, such as CXCR5 and WDFY1[71,72]; and nutrient transporters, such as Slc27a1 and Slc25a32; which may underpin rapid responses of MZ B cells to environmental changes. Interestingly, the weak effect of PDCD4 loss on global RNA translation could be explained by the decreased abundance of many factors involved in translation initiation. For example, the copy numbers of eIF4A1 were approximately 40% lower in PDCD4 KO MZ

B cells compared to WT. Similarly, the copy numbers of all eIF3 complex subunits were on average reduced by 50% (Fig. 8g). This indicates that MZ B cells downregulate the machinery for translation initiation to counterbalance the loss of PDCD4. Although the molecular mechanism remains unknown, our data demonstrate a B cell-intrinsic requirement for PDCD4 to restrain the response to a T-cell independent antigen.

## Discussion
We provide a resource that maps how the transcriptome and proteome change during maturation of mouse B cells after exit from the bone marrow. By resolving the copy number of 7560 protein groups, we defined a shared proteome of non-activated peripheral B cell subtypes. In addition, the selective remodelling of 1816 protein groups defines the identity and function of transitional and mature B cell subsets. These differences include genes encoding sensory receptors, such as TLRs and NLRs; components of metabolic pathways; and transcription factors controlling cell fate. Underscoring the utility of our resource, in many cases, the roles of these genes in determining the functional properties of B cell subsets have yet to be investigated. Furthermore, by integrative analysis of transcriptome and proteome we discovered a subset of genes characterised by the expression of full-length mRNAs which accumulate without detectable protein. We have termed these poised mRNAs.

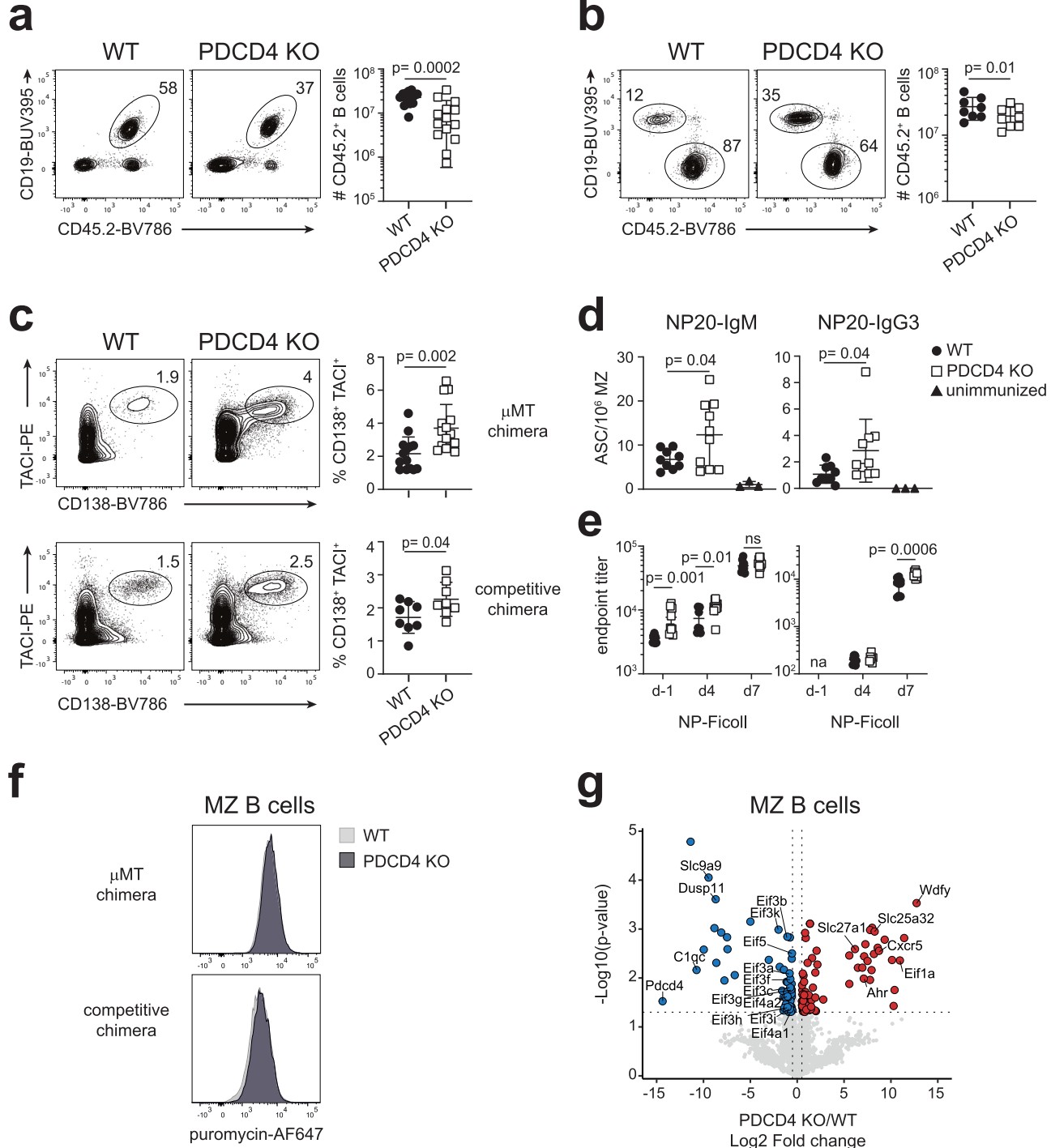

**Fig. 8 | PDCD4 restrains MZ B cell responses to NP-Ficoll immunization.**
Reconstitution of WT (filled circles) and PDCD4 KO (open squares) μMT chimeras
(**a**) or competitive chimeras (**b**) was analysed in blood samples 8-9 weeks after BM
transfer. Contour plots represent detection of transferred CD45.2⁺ cells by flow
cytometry. Graphs display numbers of CD45.2⁺ CD19⁺ B220⁺ B cells. **c** Chimeric
mice were immunized *i.v.* with NP-Ficoll and analysed after 7 days. Contour plots
and graphs depict percentage of CD138⁺ TACI⁺ CD19^int/low IgD⁻ CD45.2⁺ PBs of μMT
chimeras (top) or competitive chimeras (bottom). Full gating strategy is reported in
Supplementary Fig 9b–c. **a–c** *n* = 14–15 mice for μMT chimeras; *n* = 8 mice for
competitive chimeras. **d** ELISpot-analysis of splenic NP20-IgM (left) and NP20-IgG3
(right) antibody-secreting cells (ASC) 7 days after NP-Ficoll immunization of μMT
chimeric mice (*n* = 9–10 mice per group). Unimmunized WT μMT chimeric mice
were used as a control (*n* = 3 mice). ASC numbers were calculated as proportion of
10⁶ MZ B cells. **e** Endpoint titers of NP20-IgM (left) and NP20-IgG3 (right) were

measured in serum of μMT chimeric mice before (d-1) and after (d4, d7) NP-Ficoll
immunization (*n* = 9–10 mice per group). **a–e** Graphs display data pooled from at
least two independently performed experiments (mean ± SD). Two-tailed unpaired
Student's *t*-test analysis was performed between WT and PDCD4 KO samples.
**f** Puromycin incorporation 30 min following administration to μMT chimeras (top)
or competitive chimeras (bottom) 9 weeks after BM transfer. Overlay histograms
indicate puromycin incorporation of WT (light grey) and PDCD4 KO (dark grey) MZ
B cells. **g** Volcano plot of proteins quantified by mass spectrometry from FACS-
sorted MZ B cells that were derived from WT or PDCD4 KO competitive chimeras.
Vertical dashed lines indicate fold changes of 0.5. Horizontal dashed lines indicate *p*
values of 0.05, which were calculated using a two-tailed *t*-test with unequal variance
(*n* = 3 mice per group from two independently performed BM chimeras). Source
data are provided as Source Data file.

Our transcriptome analysis combined the deep sequencing of cDNA fragments on the Illumina platform with the full-length cDNA sequencing using nanopores. This approach validated the coding potential of poised mRNAs and distinguished them from incompletely processed transcripts or degradation intermediates. In addition, these datasets provide an opportunity to identify novel transcript isoforms expressed during B cell maturation and gain a better understanding of the differential regulation of transcripts produced from the same gene.

Poised mRNAs are emerging as a common feature for functionally "pre-arming" immune cells. They empower the innate functions of natural killer (NK) cells, NKT cells, mast cells, basophils, and eosinophils[54,55], as well as the effector function of naive and memory T cells[28,73,74]. In these contexts, however, poised mRNAs have been mainly associated with the rapid secretion of cytokines or anabolic pathways. We show here that both transitional and mature B cells express poised mRNAs, yet their nature and abundance differ between the different subsets. We specifically focused on transcripts encoding activation genes, transcription factors, and UPR mediators, thus linking poised transcripts to early activation and antibody secretion. For example, B cells express transcripts encoding CD69 and NUR77, which are known to be rapidly translated following activation. They express unspliced mRNA encoding XBP1, a key transcription factor for plasma cell differentiation that must spliced in the cytoplasm to enable the UPR[75,76]. They also express *Atf4* mRNA, the translation of which is known to be repressed by the presence of open reading frames residing 5' of the ATF4 coding region[77]. ATF4 is selectively translated when EIF2 is limiting and may promote metabolic adaptation and stress resistance[78], but its role in B cell activation and differentiation has not been established. EIF2-dependent translation inhibition can be regulated by four different kinases, HRI, PKR, PERK and GCN2 kinases. We found that PKR is 2.5-fold more abundant in MZ B cells compared to the other analysed subsets. Interestingly, PKR integrates sensing of double-stranded RNAs to NFκB and MAP kinase activation[61,79], as well as controlling translation via phosphorylation of eIF2α. Considering that MZ B cells also uniquely express TLR3, further investigation of how PKR impacts the activation of MZ B cells is warranted.

A further mechanism of regulation that we identified in B cells is mediated by PDCD4, a novel immune regulator which remains largely uncharacterised in primary cells. Germline deletion of PDCD4 promotes resistance to autoimmunity[80] and improves the anti-tumour response of CD8 T cells[81], while causing development of B cell lymphomas in old mice[80]. We found that PDCD4 restrains the response of B cells, and in part by MZ B Cells, to NP-Ficoll immunization in vivo. Whether PDCD4 controls the ability of B cells to proliferate and/or differentiate into antibody-producing plasmablasts, or the ability of plasmablasts to accumulate and/or survive remains to be investigated. Currently, the best-known function of PDCD4 is to limit translation by sequestering the RNA helicase eIF4A1[82]. The crystal structure of the eIF4A:PDCD4 complex showed that one molecule of PDCD4 binds to two eIF4A molecules[63–65,83]. MZ B cells are the only peripheral B cell subset that express PDCD4 in sufficient amounts to sequester the majority of eIF4A1. This limiting property of PDCD4 is consistent with its rapid disappearance following B cell activation[84], which would then permit rapid increases in translation. It has been shown that PDCD4 is degraded upon activation of mTORC1[84–87]. Constitutively active mTORC1 impairs MZ B cell formation and abolishes antibody-response to NP-Ficoll immunization[88]. In wild-type B cells, mTORC1 promotes the expression of UPR-related genes, thereby enhancing antibody production following activation[16]. Deconvolution of PDCD4/mTORC1 regulatory circuit and identification of the direct targets of PDCD4 in B cells might explain how translational regulation controls their ready-to-respond state.

In conclusion, our study provides a valuable resource of differentially expressed genes characteristic of the stages of peripheral B cell maturation. In addition, it offers a framework to further elucidate the role of regulation of B cell function by transcriptional and post-transcriptional mechanisms.

## Methods

### Mice and in vivo experiments

C57BL/6 (JAX stock #000664), B6.SJL-*Ptprc^aPepc^b*/BoyJ (JAX stock #002014) and B6.SJL-Ighm^tmICgn (JAX stock #002288) mice were bred and housed at the Biological Support Unit (BSU) of the Babraham Institute (UK). Since the opening of the Babraham BSU in 2009, no primary pathogens or additional agents listed in the FELASA recommendations have been confirmed during health monitoring surveys of the stock holding rooms. Ambient temperature was -19–21 °C and relative humidity 52%. Lighting was provided on a 12-hour light: 12-hour dark cycle including 15 min 'dawn' and 'dusk' periods of subdued lighting. After weaning, mice were transferred to individually ventilated cages with 1–5 mice per cage. Mice were fed CRM (P) VP diet (Special Diet Services) ad libitum and received seeds (e.g. sunflower, millet) at the time of cage-cleaning as part of their environmental enrichment. B6.129S6-Pdcd4^tmIYhcn/J mice[81] (JAX stock #018164) were bred and housed under specific pathogen-free conditions in the Central Animal Facility of the Medical School of Otto-von-Guericke-University of Magdeburg (Germany) and genotyped routinely as described[80]. All mouse experiments were approved by the Babraham Institute Animal Welfare and Ethical Review Body, and complied with existing European Union and United Kingdom Home Office legislation and local standards.

For paired proteomics and transcriptomics, 12-week-old male and female C57BL/6 mice were used. For generation of bone marrow (BM) chimeras, 8 to 12-week-old male B6.SJL-*Ptprc^aPepc^b*/BoyJ (B6.SJL-CD45.1) mice were lethally irradiated (2 × 5.0 Gy) and reconstituted with 3 × 10^6 BM cells. Donor BM cells were prepared from 9 to 10-week-old male C57BL/6, B6.129S6-Pdcd4^tmIYhcn/J, B6.SJL-*Ptprc^aPepc^b*/BoyJ and B6.SJL-Ighm^tmICgn mice. In particular, μMT BM chimeras were generated by transferring a mixture of 80% B6.SJL-Ighm^tmICgn (μMT-CD45.1) and 20% C57BL/6 (WT) or B6.129S6-Pdcd4^tmIYhcn/J (PDCD4 KO) CD45.2 cells. Competitive BM chimeras were generated by transferring a mixture of 40% B6.SJL-CD45.1 and 60% C57BL/6 (WT) or B6.129S6-Pdcd4^tmIYhcn/J (PDCD4 KO) CD45.2 cells. Prior experimental readout, mice were euthanised with cervical dislocation and pith. Peripheral blood was sampled to assess reconstitution 8–9 weeks after cell transfer. 1–2 weeks later, chimeric mice were immunized intravenously with 10 μg/ml NP-Ficoll (Biosearch Technologies) in 100 μl PBS. For in vivo puromycin incorporation assay, chimeric mice received 500μg puromycin (Sigma-Aldrich) in 200 μl PBS intraperitoneally, and were analysed 30 min later.

### Cell sorting and flow cytometry

For cell surface staining, single cell suspensions from tissues were prepared in PBS supplemented with 1% FCS and 2 mM EDTA. All cells were blocked with FcγR(CD16/32)-blocking antibody (2.4G2, BioXcell) and incubated with fixable cell viability dye eF780 (Thermofisher) to exclude dead cells from the analysis. B cell populations were sorted using anti-CD19 (6D5), anti-CD93 (AA4.1), anti-CD23 (B3B4), anti-CD21 (7G6) and anti-IgM (II/41) antibodies. Paired proteomics and transcriptomics were carried out on the following splenic B cell subsets: T1 (CD19+ CD93+ IgM+ CD23-), T2 (CD19+ CD93+ IgM+ CD23+), MZ (CD19+ CD93- CD21+ CD23-) and FoB (CD19+ CD93- CD21- CD23+) cells. Full gating strategy is reported in Supplementary Fig 9a.

For flow cytometry analysis cells were labelled for 30 min at 4 °C with the following monoclonal antibodies: anti-CD45.1 (A20), anti-CD45.2 (104), anti-CD19 (1D3 or 6D5), anti-B220 (RA3-6B2), anti-CD4 (RM4-5), anti-CD8 (53-6.7), anti-Ly6C/G (RB6-8C5), anti-CD93 (AA4.1), anti-CD23 (B3B4), anti-CD21 (7G6), anti-CD1D (1B1), anti-IgD (11–26 c.2a), anti-IgM (II/41 or RMM-1), anti-IgG3 (R40–82), anti-CD138 (281-2), anti-CD267/TACI (8F10-3), anti-CD180 (RP/14), anti-

CD40 (1C10), anti-CD22 (OX-97). Intracellular staining was performed using the CytoFix/CytoPerm kit (BD Biosciences). Puromycin was detected with an AF647-conjugated antibody (12D10, Sigma-Aldrich). eIF4A1 was detected using a rabbit polyclonal antibody (ab31217), PDCD4 using a rabbit monoclonal antibody (ab79405, both Abcam). In both cases secondary staining was performed with an AF647-conjugated donkey anti-rabbit IgG(H + L) antibody (Jackson). NP-specific cells were detected with 4-Hydroxy-3-iodo-5-nitrophenylacetic acid (NIP) conjugated to biotin through BSA followed by secondary staining with a streptavidin antibody. Full gating strategies intracellular NIP staining of PB derived from µMT and competitive chimeras are reported in Supplementary Fig 9b, c. The name, clone name, fluorochrome, catalogue number, (most used) lot number, dilution factor and manufacturer of all antibodies used in this study have been listed in the Supplementary Data 6.

Data were acquired using a LSR Fortessa Flow Cytometer equipped with 355 nm, 405 nm, 488 nm, 561 nm and 640 nm lasers (BD Biosciences) and analysed with FlowJo software (TreeStar, version 10.6.1).

### ELISA and ELISpot assay
For ELISA, serum was prepared from the blood of µMT chimeric mice. ELISA plates (Nunc Maxisorp) were coated with NP20-BSA (Biosearch Technologies) and blocked with 1% BSA in PBS. Serial dilutions of serum samples (0.1% BSA/PBS) were added and incubated overnight. NP-specific IgM and IgG3 antibodies were detected using biotinylated anti-mouse IgM- or IgG3-specific immunoglobulins followed by streptavidin-HRP (Southern Biotech) and developed with SIGMAFAST OPD tablets (Sigma-Aldrich). Absorbance values at 490 nm were determined and used to calculate endpoint titers.

For ELISpot, serial dilution of splenocyte suspensions of µMT chimeric mice were added to NP20-BSA-coated MultiScreen HA mixed cellulose ester plates (Millipore, Watford, UK) previously washed and blocked with IMDM (Sigma-Aldrich) freshly supplemented with 10% FCS, 2 mM GlutaMAX and 50 µM 2-mercaptoethanol. Upon overnight incubation, cells secreting anti-NP antibody (ASC) were visualized with HRP-conjugated anti-mouse IgM or IgG3 antibodies (Southern Biotech) followed by AEC staining Kit (Sigma-Aldrich). The numbers of ASCs were quantified using Immunospot S6 Analyzer (Cellular Technology Limited).

### Sample preparation for mass spectrometry
2.5 to $3 \times 10^6$ FACS-sorted WT B cell subsets, or $1 \times 10^6$ FACS-sorted WT and PDCD4 KO MZ B cells derived from competitive BM chimeras, were washed twice with ice-cold PBS and cell pellets were snap frozen in liquid nitrogen. Cells were lysed in 5% sodium dodecyl sulphate, 50 mM TEAB pH 8.5, 10 mM TCEP under agitation. Lysates were boiled for 5 min at 95 °C, sonicated with a BioRuptor (15 cycles of 30 sec each) and treated with 1 µl benzonase for 15 min at 37 °C. Protein yield was determined using the EZQ protein quantitation it (ThermoFisher Scientific) according to manufacturer's instructions. Lysates were then alkylated with 20 mM iodoacetamide for 1 h at RT in the dark and loaded onto S-Trap mini columns (ProtiFi). Proteins were subsequently digested with 15µg Trypsin Gold (Promega) in 50 mM ammonium bicarbonate (Sigma-Aldrich) for 1.5 h at 47 °C before peptides were eluted from columns. Eluted peptides were dried by SpeedVac and resuspended in 5% formic acid for peptide fractionation by high pH reverse-phase chromatography.

Peptides were fractionated by HPLC using a Dionex Ultimate3000 system (ThermoFisher Scientific), which consists in a 25min-multitep gradient of buffer A (10 mM ammonium formate at pH 9 in 2% acetonitrile) and buffer B (10 mM ammonium formate at pH 9 in 80% acetonitrile), at a flow rate of 0.3 ml/min. Peptides were separated in 16 fractions, which were then consolidated in 8 fractions. The fractions were subsequently dried by SpeedVac and dissolved in 5% formic acid. 1 µg was analysed for each fraction using a nanoscale C18 reverse-phase chromatography system (UltiMate 3000 RSLC nano, Thermo Scientific) coupled to an Orbitrap Q Exactive Plus mass spectrometer (Thermo Scientific), as described previously[89].

### Proteomics data analysis
For WT B cell subsets raw mass spectrometry data were processed with the MaxQuant software package version 1.6.10.43, while WT and PDCD4 KO MZ B cells were processed with the MaxQuant version 2.0.1.0. Proteins and peptides were identified using a hybrid database generated from the UniProt mouse database (available from the ProteomeXchange data repository—see data availability section). This hybrid protein database consisted of manually annotated mouse SwissProt entries, along with mouse TrEMBL entries with a manually annotated homologue within the human SwissProt database. The following variable modifications were set within MaxQuant: methionine oxidation, acetylation (N-termini), glutamine and asparagine deamidation and glutamine to pyroglutamate (for B cell subset search only). Carbamidomethylation of cysteine was set as a fixed modification. Maximum missed cleavages was set at 2, while protein and PSM false discovery rate was set at 1%. Match between runs was disabled. The dataset was then filtered to remove proteins categorized as "contaminant", "reverse" and "only identified by site" using Perseus (1.6.10.45). Copy numbers were calculated using the proteomic ruler plugin within Perseus as previously described[31]. For the proteomic analysis of B cell subsets, the accuracy of quantitation was categorized as: "high" if proteins were identified by more than eight total unique peptides and a ratio of unique + razor to total peptides ≥0.75; "medium" if proteins proteins had at least three total unique peptides and a ratio of unique + razor to total peptides ≥0.5; "low" if proteins were below these thresholds[30] (Supplementary Data 1a). Data were filtered to include only proteins identified by at least one unique peptide and in at least three out of four biological replicates. Data analysis was continued in R (4.0).

### Sample preparation for RNA-sequencing
$0.3 \times 10^6$ FACS-sorted B cell subsets were washed twice with ice-cold PBS and cell pellets were snap frozen in liquid nitrogen. RNA was extracted using the RNeasy Mini Kit (Qiagen) and its quality was assessed on a 2100 Bioanalyser (Agilent). RNA integrity numbers > 7.5 of total RNA were used to generate cDNA from polyadenylated transcripts.

For Illumina sequencing, RNA was reverse transcribed using the SMART-Seq v4 ultra low input RNA kit (Takara Bio). cDNA quality was analysed on a 2100 Bioanalyser (Agilent). mRNAseq libraries were prepared using Nextera XT DNA library preparation kit (Illumina) and quantified using KAPA library quantification kit (Roche). Barcoded libraries were multiplexed and sequenced on an Illumina HiSeq 2500-RapidRun system on a 50 bp single-end mode with a coverage of 20 M reads per sample.

For Oxford Nanopore Technologies (ONT) sequencing, libraries were prepared as described previously[90,91]. Briefly, RNA was reverse transcribed using the Smart-seq2 protocol[92], cDNA was amplified using the KAPA HiFi Uracil+ hot start polymerase mix (Roche) and PCR products were purified using 0.6X AMPure XP beads (Beckman). Equal amounts of cDNA libraries were pooled for a total of 200 fmol and sequenced with MinION R9.4.1 flow cell using the SQK-LSK109 kit on MinKNOW (21.02.1) according to the manufacturers' instructions.

### Illumina sequencing data analysis
The quality of Illumina sequencing data of B cell subsets and sequencing data retrieved from ref. 16 (GSE141419) and ref. 56 (GSE61608) was assessed using FastQC (0.11.9; http://www.bioinformatics.babraham.ac.uk/projects/fastqc/). Reads were trimmed using Trim Galore and mapped to mouse genome GRCm38 using HiSat2[93] (2.1.0). Raw counts

were calculated over mRNA features using SeqMonk (1.47.2; https://www.bioinformatics.babraham.ac.uk/projects/seqmonk/); this and subsequent analyses were performed using the GRCm38.90 annotation release. The normalised counts in transcripts per million (TPM), which corrects for transcript length and library size, were calculated using StrigTie[94] (2.1.1). When indicated, ENSEMBL annotation was used to filter for protein-coding transcripts and exclude mis-annotation of non-coding or predicted genes. Genes with TPM > 1 were considered as expressed, unless differently specified. DESeq2[95] (1.30.1) was used to calculate differential RNA abundance between two conditions and performed using default parameters, with "normal" log2 fold change shrinkage. Information on biological replicates were included in the design formula to have paired analysis.

### ONT sequencing data analysis
Basecalling of reads was performed with guppy basecaller (4.0.11) in high accuracy mode. Reads with a mean sequence quality score higher than 7 were demultiplexed with the guppy_barcoder tool and processed with pychopper (2.5.0) in order to identify, orient and trim the full-length cDNA reads. Pychopped reads were then aligned to mouse genome reference GRCm38 using minimap2 (2.17-r941)[96] with splice aware setting and analysed with FLAIR pipeline[97] for isoform identification. Aligned SAM files from minimap2 were converted to BAM format after sorting and indexing with samtools (1.9). BAM files for replicate datasets were merged using samtools (1.9) and visualised with the Integrative Genomics Viewer (IGV 2.7.2) in expanded display mode for aligned read tracks.

### Transcriptomics and proteomics data integration
To combine transcriptomic and proteomic data, Ensembl ID of corresponding UniProt ID were retrieved using Perseus (1.6.10.45), Ensembl BioMart (2.46.3) and manually cured for obvious mis-annotations. Protein copy numbers of all four biological replicates were averaged and integrated with average mRNA abundance in TPM. Pearson correlation coefficient ($r$) and Spearman correlation coefficient ($\rho$) were calculated in R (4.0) using average RNA in log10 TPM and average protein in log10 copy numbers per B cell subset. To generate a trend line describing the relationship between mRNA TPM and protein copy numbers per B cell subset, ggplot2::geom_density2d was used to plot density contours including only genes that were detected by both RNA-seq and proteomics, and ggplot_build to extract their coordinates. Assuming a positive relationship between mRNA and protein expression level, we identified the coordinates for each contour that gave the maximum or minimum sum of x (TPM) and y (protein copies) and used these in a linear regression model to plot a best fit line describing each B cell population. $R^2$ values for these ranged between 0.95–0.968.

To test whether detection of proteins differed from what would be expected by chance for the early activation and the plasma cell-related gene sets, control gene sets were generated by matching expression levels (log2 TPM) of protein-coding mRNAs (i.e. with a consensus coding sequence) to each gene set of interest. In particular, for each gene within the early activation or the plasma cell-related gene sets, 100 genes with the closest expression in a given B cell population were identified. One of these was then randomly selected and used as a corresponding control. By repeating this process 100 times, we generated 100 control sets for each gene set and cell type. The median, 5th and 95th percentile of the number of genes that were detected at the protein level, across the 100 control sets, were then calculated and compared to the number of genes for which protein was detected in the early activation or the plasma cell-related gene sets.

For visualization of the expected range of protein abundance for each individual poised gene, the 100 protein-coding genes closest in RNA expression were selected, regardless of whether they were also selected for another gene, and without random selection.

### Statistics and data visualization
For in vivo mouse experiments, statistical analysis was performed with GraphPad Prism 9 using two-tailed unpaired Student $t$-test when comparing two groups. $P$ values < 0.05 were considered statistically significant.

For statistical analysis of proteomic data, protein copy numbers were normalized for the total sum of protein copy numbers per sample and log2 transformed. Intergroup differences were calculated in R using ANOVA test followed by a Benjamini-Hochberg multiple testing correction. Statistics of RNA-sequencing data was conducted in DESeq2 (1.30.1), which uses the Wald test followed by a Benjamini-Hochberg multiple testing correction. For both proteomic and transcriptomic analysis, differences were considered significant if adjusted $p$-value (FDR) was <0.05. Plots were generated with ggplot2 (3.3.3) and GraphPad Prism 9, venn diagrams with VennDiagram CRAN package (1.6.20), heat maps with pHeatmap (1.0.12).

### Reporting summary
Further information on research design is available in the Nature Portfolio Reporting Summary linked to this article.

## Data availability
All data generated and analyzed in this study are included in this article, its supplementary information, or have been made available in public repositories. Raw mass spectrometry data files, MaxQuant analysis files and fasta database files are available from the ProteomeXchange data repository. The proteomic data of T1, T2, MZ and FoB cells are accessible with the identifier PXD043349, whereas the proteomic data of WT and PDCD4 KO MZ B cells are accessible with the identifier PXD043351. Analysed proteomics data used to generate figures are available in Supplementary Data 1a–c. Both Illumina and ONT sequencing data generated in this study are available from the NCBI Gene Expression Omnibus (GEO) repository under the accession code GSE178728. Calculated TPMs and protein copy numbers of genes identified by Illumina sequencing and proteomics are reported in Supplementary Data 2. Lists of early activation genes and PB-related genes extracted upon DESeq2 analysis of RNA-sequencing from refs. 16,56 are reported in Supplementary Data 3a, b. A side-by-side comparison of TPMs of genes that were detected by both ONT and Illumina sequencing is reported in Supplementary Data 4. A list of poised mRNAs in B cells is provided in Supplementary Data 5a, b. Flow cytometry data that support the finding of this study are available from the authors upon request. Source data are provided with this paper.

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

## Acknowledgements

We thank Doreen Cantrell for support and the Biological Support Unit, Next Generation Sequencing, Flow Cytometry and Bioinformatics facilities of the Babraham Institute, the Mass Spectrometry facility of University of Dundee, and the Next Generation Sequencing of the CRUK for outstanding technical assistance. We thank Simon Andrews, Claudia Ribeiro de Almeida and Sarah E. Bell for critical review of the manuscript, and Sarah Collinson for formatting the Supplementary Figure File. This study was funded by the Biotechnology and Biological Sciences Research Council (BBSRC) (BBS/E/B/000C0427; BBS/E/B/000C0428), the BBSRC Core Capability Grant to the Babraham Institute, the H2020-Marie Skłodowska-Curie Innovative Training Network 765158 "COSMIC" and a Wellcome Investigator award (200823/Z/16/Z). F.S. was supported by European Molecular Biology Organization (EMBO) Long-Term Fellowship (ALTF 880-2018) and H2020-Marie Skłodowska-Curie Individual Fellowship (841930, B-different). M.C.B.W. was supported by Deutsche Forschungsgemeinschaft (DFG) SFB854 B14 and Br1860/12.

## Author contributions

F.S: conceptualization; methodology; investigation; formal analysis; visualization; data curation; writing—original draft preparation—review and editing; project administration; funding acquisition. A.J.M.H: conceptualization; methodology; formal analysis; writing—review and editing. L.S.M: conceptualization; formal analysis; visualization; data curation; writing—review and editing. O.G: methodology; formal analysis; visualization; data curation; writing—review and editing. M.S: validation; writing—review and editing. H.L. and M.C.B.W: resources; writing—review and editing. M.T: conceptualization; writing—review and editing; project administration; funding acquisition; supervision.

## Competing interests

The authors declare no competing interests.
