## [Peer Review File · Nature Communications]

An integrated proteome and transcriptome of B cell maturation defines poised activation states of transitional and mature B cellsEditorial Note: This manuscript has been previously reviewed at another journal that is not operating a transparent peer review scheme. This document only contains reviewer comments and rebuttal letters for versions considered at *Nature Communications*.

REVIEWERS' COMMENTS

Reviewer #1 (Remarks to the Author):

The authors have addressed my key concerns of the last round of review about high variations among LFQ biological replicates. I therefore recommend its publication on Nature Communication.

Reviewer #4 (Remarks to the Author):

I have been asked to come in as a mediator for this manuscript by Salerno et al "An integrated proteome and transcriptome of B cell maturation defines poised activation states of transitional and mature B cells". This manuscript was reviewed at Nature Immunology and has now been transferred to Nature Communications and reviewer #3 is no longer available to re-review the revised manuscript.

The strengths are in the data set being a valuable resource with interest from a lot of researchers. The weakness, which has already been pointed out by other reviewers is the high variation in the data between repeated measures". I do agree it is a big ask and not feasible for the authors to repeat the experiments using a different approach. A lot of the paper shows differences in protein expression without verification that these differences results in mechanisms that explain the biological differences in these B cell subsets. For instance, it would have been nice to see functional readouts to verify some of the proteomics results ie response of B cell subsets (T1, T2, MZ, FoB) to cytokines that signal via STAT1, STAT2 and STAT4 ie IFNs and IL-12. Similarly, activation via TLR3, TLR7 and TLR9, latter being not particularly different.

In summary if the aim is to generate a publicly available proteome-wide resource that allows users to identify quantitatively differentially expressed proteins in transitional and mature B cell subsets, then I believe this has been achieved.

Reviewer #1 (Remarks to the Author):

The authors have addressed my key concerns of the last round of review about high variations among LFQ biological replicates. I therefore recommend its publication on Nature Communication.

We thank reviewer #1 for their contribution to improve our work and their positive response.

Reviewer #4 (Remarks to the Author):

I have been asked to come in as a mediator for this manuscript by Salerno et al “An integrated proteome and transcriptome of B cell maturation defines poised activation states of transitional and mature B cells”. This manuscript was reviewed at Nature Immunology and has now been transferred to Nature Communications and reviewer #3 is no longer available to re-review the revised manuscript.

The strengths are in the data set being a valuable resource with interest from a lot of researchers. The weakness, which has already been pointed out by other reviewers is the high variation in the data between repeated measures”. I do agree it is a big ask and not feasible for the authors to repeat the experiments using a different approach. A lot of the paper shows differences in protein expression without verification that these differences results in mechanisms that explain the biological differences in these B cell subsets. For instance, it would have been nice to see functional readouts to verify some of the proteomics results ie response of B cell subsets (T1, T2, MZ, FoB) to cytokines that signal via STAT1, STAT2 and STAT4 ie IFNs and IL-12. Similarly, activation via TLR3, TLR7 and TLR9, latter being not particularly different.

In summary if the aim is to generate a publicly available proteome-wide resource that allows users to identify quantitatively differentially expressed proteins in transitional and mature B cell subsets, then I believe this has been achieved.

We thank reviewer #4 for their review our manuscript and for recognizing the value and broad interest of our work. As reviewer #4 has pointed out, our aim was to generate a publicly available proteome-wide resource that can be interrogated by other scientists and inspire new lines of investigation.

In this manuscript we have validated protein expression of key factors by flow cytometry (please see Figs 2, 6, S1, S3) and further investigated the function of PDCD4 as an exemplar of the predictive power of our resource. We selected PDCD4 because it is a novel regulators of B cell function. We agree with reviewer #4 that the differential expression of STAT proteins in B cell subsets and the selective expression of TLR3 and TLR7 in MZ B cells is of interest. We suggest that a detailed functional characterisation and pursuit of mechanistic insight into the responses of B cells to signalling by these receptors and STAT-transcription factors, including their target genes in B cells, is a significant body of work and worthy of a follow on study in its own right.